# Likelihood-Free Overcomplete ICA and Applications in Causal Discovery

**Chenwei Ding**
UBTECH Sydney AI Centre
School of Computer Science, Faculty of Engineering
University of Sydney
cdin2224@uni.sydney.edu.au

**Mingming Gong**
School of Mathematics and Statistics
University of Melbourne
mingming.gong@unimelb.edu.au

**Kun Zhang**
Department of Philosophy
Carnegie Mellon University
kunz1@cmu.edu

**Dacheng Tao**
UBTECH Sydney AI Centre
School of Computer Science, Faculty of Engineering
University of Sydney
dacheng.tao@uni.sydney.edu.au

## Abstract

Causal discovery witnessed significant progress over the past decades. In particular, many recent causal discovery methods make use of independent, non-Gaussian noise to achieve identifiability of the causal models. Existence of hidden direct common causes, or confounders, generally makes causal discovery more difficult; whenever they are present, the corresponding causal discovery algorithms can be seen as extensions of overcomplete independent component analysis (OICA). However, existing OICA algorithms usually make strong parametric assumptions on the distribution of independent components, which may be violated on real data, leading to sub-optimal or even wrong solutions. In addition, existing OICA algorithms rely on the Expectation Maximization (EM) procedure that requires computationally expensive inference of the posterior distribution of independent components. To tackle these problems, we present a Likelihood-Free Overcomplete ICA algorithm (LFOICA[1]) that estimates the mixing matrix directly by back-propagation without any explicit assumptions on the density function of independent components. Thanks to its computational efficiency, the proposed method makes a number of causal discovery procedures much more practically feasible. For illustrative purposes, we demonstrate the computational efficiency and efficacy of our method in two causal discovery tasks on both synthetic and real data.

## 1  Introduction

Discovering causal relations among variables has been an important problem in various fields such as medical science and social sciences. Because conducting randomized controlled trials is usually expensive or infeasible, discovering causal relations from observational data, *i.e.,*causal discovery

[1, 2]) has received much attention in the past decades. Classical causal discovery methods, such as PC [2] and GES [3], output multiple causal graphs in the Markov equivalence classes. Since the seminal work [4], there have been various methods that have complete identifiability of the causal structure by making use of constrained Functional Causal Models (FCMs), such as linear non-Gaussian models [4], nonlinear additive model [5], and post-nonlinear model [6]. Some recent researches also consider the heterogeneous case [7, 8, 9, 10, 11].

Whenever there are essentially unobservable direct common causes of two variables (known as confounders), causal discovery can be viewed as learning with hidden variables. With the linearity and non-Gaussian noise constraints, it has been shown that the causal model is even identifiable from data with measurement error [12] or missing common causes [13, 14, 15, 16, 17]. The corresponding causal discovery algorithms can be seen as extension of overcomplete independent component analysis (OICA). Unlike regular ICA [18], in which the mixing matrix is invertible, OICA cannot utilize the change of variables technique to derive the joint probability density function of the data, which is a product of the densities of the independent components (ICs), divided by some value depending on the mixing matrix. The joint density immediately gives rise to the likelihood.

To perform maximum likelihood learning, exisiting OICA algorithms typically assume a parametric distribution for the hidden ICs. For example, if assuming each IC follows a Mixture of Gaussian (MoG) distribution, we can simply derive the likelihood for the observed data. However, the number of Gaussian mixtures increases exponentially in the number of ICs, which poses significant computational challenges. Many of existing OICA algorithms rely on the Expectation-Maximization (EM) procedure combined with approximate inference techniques, such as Gibbs sampling [19] and mean-field approximation [20], which usually sacrifice the estimation accuracy. Furthermore, the extended OICA algorithms for causal discovery are mostly noiseless OICA because they usually model all the noises as ICs [12, 15]. In order to apply EM, a very low variance Gaussian noise is usually added to the noiseless OICA model, resulting in very slow convergence [21]. Finally, the parametric assumptions on the ICs might be restrictive for many real-world applications.

To tackle these problems, we propose a Likelihood-Free OICA (LFOICA) algorithm that makes no explicit assumptions on the density functions of the ICs. In light of recent work on adversarial learning [22], LFOICA utilizes neural networks to learn the distribution of independent components implicitly. By minimizing appropriate distributional distance between the generated data from LFOICA model and the observed data, all parameters including the mixing matrix and noise learning network parameters in LFOICA can be estimated very efficiently via stochastic gradient descent (SGD) [23, 24], without the need to formulate the likelihood function.

Although both our work and [25] use a GAN style approach to solve ICA, they are largely different to each other. First, the main purpose of [25] is to recover the ICs instead of how the ICs are mixed (*i.e.*, the mixing matrix). It models the mixing and unmixing procedure implicitly with an encoder-decoder architecture. As a consequence of non-linearity, there is no guarantee for identifiability. In contrast, we concentrate on the mixing matrix estimation for causal discovery purpose. Second, the encoder-decoder architecture in [25] cannot be easily extended for OICA because the posterior of ICs cannot be modeled by a deterministic encoder. Third, the adversarial training target of LFOICA and [25] are different. While [25] aims at matching the joint distribution and product of marginal distribution of the recovered ICs (this is also how [25] makes the components independent), LFOICA is trained to match the distributions of the generated mixtures and true mixtures. And the estimated ICs by LFOICA are naturally independent because they are generated from independent latent noises with separate networks.

The proposed LFOICA will make a number of causal discovery procedures much more practically feasible. For illustrative purposes, we extend our LFOICA method to tackle two causal discovery tasks, including causal discovery from data with measurement noise [12] and causal discovery from low-resolution time series [15, 16]. Experimental results on both synthetic and real data demonstrate the efficacy and efficiency of our proposed method.

## 2 Likelihood-Free Over-complete ICA

### 2.1 General Framework

Linear ICA assumes the following data generation model:

$$\mathbf{x} = \mathbf{As}, \tag{1}$$

where $\mathbf{x} \in \mathbb{R}^p, \mathbf{s} \in \mathbb{R}^d, \mathbf{A} \in \mathbb{R}^{p \times d}$ are known as mixtures, independent components (ICs), and mixing matrix respectively. The elements in $\mathbf{s}$ are supposed to be independent from each other and each follows a non-Gaussian distribution (or at most one of them is Gaussian). The goal of ICA is to recover both $\mathbf{A}$ and $\mathbf{s}$ from observed mixtures $\mathbf{x}$. However, in the context of causal discovery, our main goal is to recover a constrained $\mathbf{A}$ matrix. When $d > p$, the problem is known as overcomplete ICA (OICA).

In light of recent advances in Generative Adversarial Nets (GANs) [22], we propose to learn the mixing matrix in the OICA model by designing a generator that allows us to draw samples easily. We model the distribution of each source $s_i$ by a function model $f_{\theta_i}$ that transforms a Gaussian variable $z_i$ to the non-Gaussian source. More specifically, the $i$-th source can be generated by $\hat{s}_i = f_{\theta_i}(z_i)$, where $z_i \sim \mathcal{N}(0, 1)$. Thus, the whole generator that generate $\mathbf{x}$ can be written as

$$\hat{\mathbf{x}} = \mathbf{A}[\hat{s}_1, \ldots, \hat{s}_d]^\intercal = \mathbf{A}[f_{\theta_1}(z_1), \ldots, f_{\theta_d}(z_d)]^\intercal = G_{\mathbf{A}, \boldsymbol{\theta}}(\mathbf{z}), \tag{2}$$

where $\boldsymbol{\theta} = [\theta_1, \ldots, \theta_d]^\intercal$ and $\mathbf{z} = [z_1, \ldots, z_d]^\intercal$. Figure 1 shows the graphical structure of our LFOICA generator $G_{\mathbf{A}, \boldsymbol{\theta}}$ with 4 sources and 3 mixtures. We use a multi-layer perceptron (MLP) to model each $f_{\theta_i}$. While most of the previous algorithms for both overdetermined [26, 25, 27, 28] and overcomplete [29] scenarios try to minimized the dependence among the recovered components, the components $\hat{s}_i$ recovered by LFOICA are essentially independent because the noises $z_i$ are independent, according to the generating process.

The LFOICA generator $G_{\mathbf{A}, \boldsymbol{\theta}}$ can be learned by minimizing the distributional distance between the data sampled from the generator and the observed $\mathbf{x}$ data. Various distributional distances have been applied in training generative networks, including the Jensen-Shannon divergence [22], Wasserstein distance [30], and Maximum Mean Discrepancy (MMD) [31, 32]. Here we adopt MMD as the distributional distance as it does not require an explicit discriminator network, which simplifies the whole optimization procedure. Specifically, we learn the parameters $\boldsymbol{\theta}$ and $\mathbf{A}$ in the generator by solving the following optimization problem:

$$
\begin{aligned}
\mathbf{A}^*, \boldsymbol{\theta}^* &= \arg\min_{\mathbf{A}, \boldsymbol{\theta}} M\left(\mathbb{P}(\mathbf{x}), \mathbb{P}(G_{\mathbf{A}, \boldsymbol{\theta}}(\mathbf{z}))\right) \\
&= \arg\min_{\mathbf{A}, \boldsymbol{\theta}} \left\| \mathbb{E}_{\mathbf{x} \sim p(\mathbf{x})}[\phi(\mathbf{x})] - \mathbb{E}_{\mathbf{z} \sim p(\mathbf{z})}[\phi(G_{\mathbf{A}, \boldsymbol{\theta}}(\mathbf{z}))] \right\|^2,
\end{aligned}
\tag{3}
$$

where $\phi$ is the feature map of a kernel function $k(\cdot, \cdot)$. MMD can be calculated by using kernel trick without the need for an explicit $\phi$. By choosing characteristic kernels, such as Gaussian kernel, MMD is guaranteed to match the distributions [33]. In practice, we optimize some empirical estimator of (3) on minibatches by stochastic gradient descent (SGD). The entire procedure is shown in Algorithm 1.

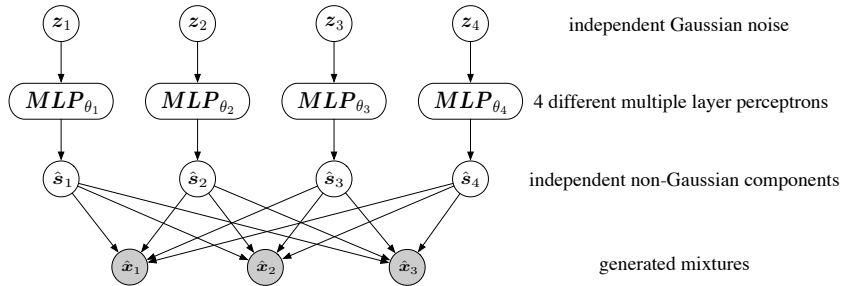

Figure 1: generator architecture of LFOICA. $z_1, z_2, z_3, z_4$ are i.i.d Gaussian noise variables.

The identifiability of the mixing matrix $\mathbf{A}$ in our model ($\mathbf{x} = G_{\mathbf{A}, \boldsymbol{\theta}}(\mathbf{z}) = \mathbf{A}[f_{\theta_1}(z_1), \ldots, f_{\theta_d}(z_d)]^\intercal$) follows the identifiability results for OICA [34], which is summarized in the following theorem.

---

**Algorithm 1** Likelihood-Free Overcomplete ICA (LFOICA) Algorithm

---
1: Get a minibatch of i.i.d samples $\mathbf{z}$ from Gaussian noise distribution.
2: Generate mixtures using (2).
3: Get a minibatch of samples from the distribution of observed mixtures $p(\mathbf{x})$.
4: Update $\mathbf{A}$ and $\boldsymbol{\theta}$ by minimizing the empirical estimate of (3) on the minibatch.
5: Repeat step 1 to step 4 until max iterations reached.

---

**Theorem 1** *Given two OICA models* $\mathbf{x} = G_{\mathbf{A},\boldsymbol{\theta}}(\mathbf{z})$ *and* $\mathbf{x}' = G_{\mathbf{A}',\boldsymbol{\theta}'}(\mathbf{z}')$ *that specify distributions* $\mathbb{P}(\mathbf{x})$ *and* $\mathbb{P}(\mathbf{x}')$*, respectively. Under the non-Gaussian assumption of* $f_i(\mathbf{z}_i)$ *(please refer to Theorem 1 & 3 in [34] for precise definitions), if* $MMD(\mathbb{P}(\mathbf{x}), \mathbb{P}(\mathbf{x}')) = 0$*, then* $\mathbf{A}' = \mathbf{A}\mathbf{P}_p\mathbf{S}_p$*, where* $\mathbf{P}_p$ *is a* $p \times p$ *column permutation matrix and* $\mathbf{S}_p$ *is a* $p \times p$ *scaling matrix.*

The proof is almost the same as that of Theorem 3 in [34], except that in order to guarantee $\mathbb{P}(\mathbf{x}) = \mathbb{P}(\mathbf{x}')$, we use $MMD = 0$ while [34] uses maximum likelihood (KL divergence). Given the identifiability results, the estimated mixing matrix converges to the scaled and permuted version of the true mixing matrix and so do the source distributions. The parameters in our MLPs (*i.e.*,$\boldsymbol{\theta}$) are not identifiable ($\boldsymbol{\theta} \neq \boldsymbol{\theta}'$), but we do not need the identifiability of $\boldsymbol{\theta}$ to perform certain tasks, such as the two causal discovery tasks studied in this paper.

## 2.2 Practical Considerations

We consider two important issues when applying LFOICA to real applications.

**Sparsity**  Based on the fact that the mixing matrix is sparse in many real systems, we add a LASSO regularizer [35] to (3), resulting in the loss function $M\left(\mathbb{P}(\mathbf{x}), \mathbb{P}(G_{\mathbf{A},\boldsymbol{\theta}}(\mathbf{z}))\right) + \lambda \sum_i \sum_j |\mathbf{A}_{ij}|$. We use the stochastic proximal gradient method [36] to train our model. The proximal mapping for LASSO regularizer corresponds to the soft-thresholding operator:

$$\text{prox}_\gamma(\mathbf{A}) = S_{\lambda\gamma}(\mathbf{A}) = \begin{cases} \mathbf{A} - \lambda\gamma & \text{if } \mathbf{A} > \lambda\gamma \\ 0 & \text{if } -\lambda\gamma \leq \mathbf{A} \leq \lambda\gamma \\ \mathbf{A} + \lambda\gamma & \text{if } \mathbf{A} < -\lambda\gamma \end{cases},$$

where $\lambda, \gamma$ are the regularization weight and the learning rate, respectively. The soft-thresholding operator is applied after each gradient descent step:

$$\mathbf{A}^{(t)} = \text{prox}_{\lambda\gamma_t}\left(\mathbf{A}^{(t-1)} - \gamma_t \nabla M_{\mathbf{A}^{(t-1)}}(\cdot)\right), \quad t = 1, 2, 3, \ldots .$$

**Insufficient data**  When we have rather small datasets, it is beneficial to have certain "parametric" assumptions on the source distributions. Here we use Mixture of Gaussian (MoG) distribution to model the non-Gaussian distribution of independent components. Specifically, the distribution for the $i$-th IC is

$$p_{\hat{s}_i} = \sum_{j=1}^m P(z_i = j) P(\hat{s}_i | z_i = j) = \sum_{j=1}^m w_{i,j} \mathcal{N}\left(\hat{s}_i | \mu_{i,j}, \sigma_{i,j}^2\right), \quad i = 1, 2, \ldots, d,$$

where $m$ is the number of Gaussian components in MoG and $w_{ij}$ is the mixture proportions satisfying $\sum_{j=1}^m w_{ij} = 1$. If we do not wish to learn $w_{ij}$, we can first sample $z_i$ from the categorical distribution $P(z_i = j) = w_{ij}$, and then use the reparameterization trick in [37] to sample from $P(\hat{s}_i | z_i)$ by an encoder network $\hat{s}_i = \mu_{i,z_i} + \epsilon\sigma_{i,z_i}$, where $\epsilon \sim \mathcal{N}(0,1)$. In this way, the gradients can be backpropagated to $\mu_{ij}$ and $\sigma_{ij}$. Learning $w_{ij}$ is relatively hard because $z_i$ is discrete and thus does not allow for backpropagation to $w_{ij}$. To address this problem, we adopt the Gumbel-`softmax` trick [38, 39] to sample $z_i$. Specifically, we use the following `softmax` function to generate one-hot $\tilde{\mathbf{z}}_i$:

$$\tilde{z}_{ij} = \frac{\exp\left(\left(\log\left(w_{ij}\right) + g_j\right)/\tau\right)}{\sum_{k=1}^m \exp\left(\left(\log\left(w_{ik}\right) + g_k\right)/\tau\right)}, \tag{4}$$

where $g_1, \ldots, g_m$ are i.i.d samples drawn from Gumbel (0,1), and $\tau$ is the temperature parameter that controls the approximation accuracy of `softmax` to `argmax`. By leveraging the two tricks, we can sample $\hat{s}_i$ from the generator $\hat{s}_i = \mathbf{u}\tilde{\mathbf{z}}_i + \epsilon\mathbf{v}\tilde{\mathbf{z}}_i$, where $\mathbf{u} = [\mu_{i1}, \ldots, \mu_{im}]$ and $\mathbf{v} = [\sigma_{i1}, \ldots, \sigma_{im}]$, which enables learning of all the parameters in the MoG model.

# 3 Applications in Causal Discovery

## 3.1 Causal Discovery under Measurement Error

Measurement error (*e.g.,* noise caused by sensors) in the observed data can lead to wrong result of various causal discovery methods. Recently, it was proven that the causal structure is identifiable from data with measurement error, under the assumption of linear relations and non-Gaussian noise [12]. Based on the identifiability theory in [12], we propose a causal discovery algorithm by extending LFOICA with additional constraints.

Following [12], we use the LiNGAM model [4] to represent the causal relations on the data without measurement error. More specifically, the causal model is $\tilde{\mathbf{X}} = \mathbf{B}\tilde{\mathbf{X}} + \tilde{\mathbf{E}}$, where $\tilde{\mathbf{X}}$ is the vector of the variables without measurement error, $\tilde{\mathbf{E}}$ is the vector of independent non-Gaussian noise terms, and $\mathbf{B}$ is the corresponding causal adjacency matrix in which $B_{ij}$ is the coefficient of the direct causal influence from $\tilde{X}_j$ to $\tilde{X}_i$ and $B_{ii} = 0$ (no self-influence). In fact, $\tilde{\mathbf{X}}$ is a linear transformation of the noise term $\tilde{\mathbf{E}}$ because the linear model can be rewritten as $\tilde{\mathbf{X}} = (\mathbf{I} - \mathbf{B})^{-1}\tilde{\mathbf{E}}$. Then, the model with measurement error $\mathbf{E}$ can be written as

$$\mathbf{X} = \tilde{\mathbf{X}} + \mathbf{E} = (\mathbf{I} - \mathbf{B})^{-1}\tilde{\mathbf{E}} + \mathbf{E} = \begin{bmatrix}(\mathbf{I} - \mathbf{B})^{-1} & \mathbf{I}\end{bmatrix} \begin{bmatrix} \tilde{\mathbf{E}} \\ \mathbf{E} \end{bmatrix}, \tag{5}$$

where $\mathbf{X}$ is the vector of observable variables, and $\mathbf{E}$ the vector of measurement error terms. Obviously, (5) is a special OCIA model with $\begin{bmatrix}(\mathbf{I} - \mathbf{B})^{-1} & \mathbf{I}\end{bmatrix}$ as the mixing matrix. Therefore, we can readily extend our LFOICA algorithm to estimate the causal adjacency matrix $\mathbf{B}$.

## 3.2 Causal Discovery from Subsampled Time Series

Granger causal analysis has been shown to be sensitive to temporal frequency/resolution of time series. If the temporal frequency is lower than the underlying causal frequency, it is generally difficult to discover the high-frequency causal relations. Recently, it has been shown that the high-frequency causal relations are identifiable from subsampled low-frequency time series under the linearity and non-Gaussianity assumptions [15]. The corresponding model can also be viewed as extensions of OICA and the model parameters are estimated in the (variational) Expectation Maximization framework [15]. However, with the non-Gaussian ICs, *e.g.,* MoG is used in [15], the EM algorithm is generally intractable while the variational EM algorithm loses accuracy. To make causal discovery from subsampled time series practically feasible, we further extend our LFOICA to discover causal relations from such data.

Following [15], we assume that data at the original causal frequency follow a first-order vector autoregressive process (VAR(1)):

$$\mathbf{x}_t = \mathbf{C}\mathbf{x}_{t-1} + \mathbf{e}_t, \tag{6}$$

where $\mathbf{x}_t \in \mathbb{R}^n$ is the high frequency data and $\mathbf{e}_t \in \mathbb{R}^n$ represents independent non-Gaussian noise in the causal system. $\mathbf{C} \in \mathbb{R}^{n \times n}$ is the causal transition matrix at true causal frequency with $C_{ij}$ representing the temporal causal influence from variable $j$ to variable $i$. As done in [15], we consider the following subsampling scheme under which the low frequency data can be obtained: for every $k$ consecutive data points, one is kept and the others being dropped. Then the observed subsampled data with subsampling factor $k$ admits the following representation [15]:

$$\tilde{\mathbf{x}}_{t+1} = \mathbf{C}^k\tilde{\mathbf{x}}_t + \mathbf{L}\tilde{\mathbf{e}}_{t+1}, \tag{7}$$

where $\tilde{\mathbf{x}}_t \in \mathbb{R}^n$ is the observed data subsampled from $\mathbf{x}_t$, $\mathbf{L} = [\mathbf{I}, \mathbf{C}, \mathbf{C}^2, ..., \mathbf{C}^{k-1}]$, and $\tilde{\mathbf{e}}_{\mathbf{t}} = (\mathbf{e}_{1+tk-0}^\mathsf{T}, \mathbf{e}_{1+tk-1}^\mathsf{T}, ..., \mathbf{e}_{1+tk-(k-1)}^\mathsf{T})^\mathsf{T} \in \mathbb{R}^{nk}$ is a vector containing $nk$ independent noise terms. We are interested in estimating the transition matrix $\mathbf{C}$ from the subsampled data. A graphical representation of the subsampled data is given in Figure 2(a). Apparently, (7) extends the OICA model by considering temporal relations between observed $\tilde{\mathbf{x}}_t$.

To apply our LFOICA to this problem, we propose to model the conditional distribution $\mathbb{P}(\tilde{\mathbf{x}}_{t+1}|\tilde{\mathbf{x}}_t)$ using the following model:

$$\hat{\tilde{\mathbf{x}}}_{t+1} = G_{\mathbf{C},\boldsymbol{\theta}}(\tilde{\mathbf{x}}_t, \mathbf{z}_{t+1}) = \mathbf{C}^k\tilde{\mathbf{x}}_t + \mathbf{L}[f_{\theta_1}(z_{t+1,1}), \dots, f_{\theta_{nk}}(z_{t+1,nk})]^\mathsf{T}, \tag{8}$$

which belongs to the broad class of conditional Generative Adversarial Nets (cGANs) [40]. We call this extension of LFOICA as LFOICA-conditional. A graphical representation of (8) is shown in Figure 2(b). To learn the parameters in (8), we minimize the MMD between the joint distributions of true and generated data:

$$\mathbf{C}^*, \boldsymbol{\theta}^* = \underset{\mathbf{C}, \boldsymbol{\theta}}{\arg\min} \, M\left(\mathbb{P}(\tilde{\mathbf{x}}_t, \tilde{\mathbf{x}}_{t+1}), \mathbb{P}(G_{\mathbf{C}, \boldsymbol{\theta}}(\tilde{\mathbf{x}}_t, \mathbf{z}_{t+1}), \tilde{\mathbf{x}}_{t+1})\right)$$

$$= \underset{\mathbf{C}, \boldsymbol{\theta}}{\arg\min} \, \left\| \mathbb{E}_{(\tilde{\mathbf{x}}_t, \tilde{\mathbf{x}}_{t+1}) \sim p(\tilde{\mathbf{x}}_t, \tilde{\mathbf{x}}_{t+1})}[\phi\left(\tilde{\mathbf{x}}_t\right) \otimes \phi\left(\tilde{\mathbf{x}}_{t+1}\right)] \right.$$

$$\left. - \mathbb{E}_{\tilde{\mathbf{x}}_t \sim p(\tilde{\mathbf{x}}_t), \mathbf{z}_{t+1} \sim p(\mathbf{z}_{t+1})}[\phi(\tilde{\mathbf{x}}_t) \otimes \phi\left(G_{\mathbf{C}, \boldsymbol{\theta}}(\mathbf{z}_{t+1})\right)] \right\|^2, \tag{9}$$

where $\otimes$ denotes tensor product. The empirical estimate of (9) can be obtained by randomly sampling $(\tilde{\mathbf{x}}_t, \tilde{\mathbf{x}}_{t+1})$ pairs from true data and sampling from $\mathbb{P}(\mathbf{z}_{t+1})$. Again, we can use the mini-batch SGD algorithm to learn the model parameters efficiently.

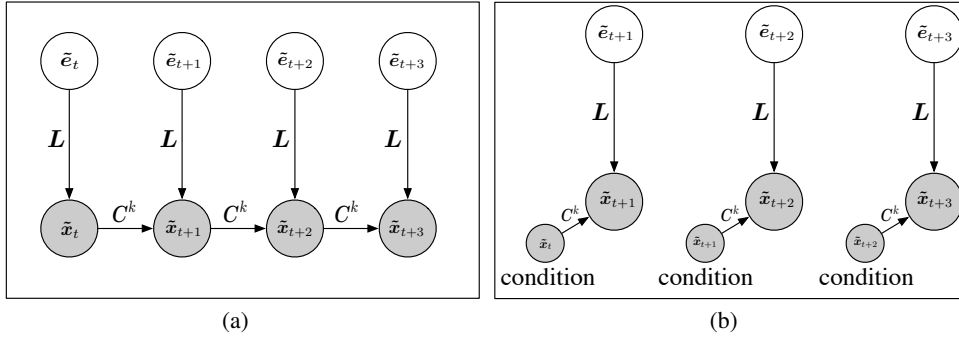

(a)                                                      (b)

Figure 2: (a) Subsampled data with subsampling factor $k$. (b) LFOICA-conditional model for subsampled data.

## 4 Experiment

In this section, we conduct empirical studies on both synthetic and real data to show the effectiveness of our LFOICA algorithm and its extensions to solve causal discovery problems. We first compare the results obtained by LFOICA and several OICA algorithms on synthetic over-complete mixtures data. Then we apply the extensions of LFOICA mentioned in Section 3.1 and 3.2 in two causal discovery problems using both synthetic and real data.

### 4.1 Recovering Mixing Matrix from Synthetic OICA Data

We compare LFOICA with several well-known OICA algorithms on synthetic OICA data.

According to [34], the mixing matrix in OICA can be estimated up to the permutation and scaling indeterminacies (including the sign indeterminacy) of the columns. However, these indeterminacies stop us from comparing the estimated mixing matrices by different OICA algorithms. In order to make the comparison achievable, we need to eliminate these indetermincies. To eliminate the permutation indetermincy, we make the non-Gaussian distribution for each synthetic IC not only independent, but also different. With different distributions for each IC, it is convenient to permute the columns to the same order for all the algorithms according to the recovered distribution of each IC. We use Laplace distributions with different variance for each IC. In order to eliminate the scaling indeterminacy, both ground-truth and estimated mixing matrix are normalized to make the L2 norm of the first column equal to 1. With the permutation and scaling indeterminacy eliminated, we can conveniently compare the mixing matrices obtained by different algorithms. To further avoid local optimum, the mixing matrix is initialized by it's true value added with noise.

Table 1 compares the mean square error (MSE) between the ground-truth mixing matrix used to generate the data and the estimated mixing matrices by different OICA algorithms. In the table, RICA represents reconstruction ICA [29], MFICA_Gauss and MFICA_MoG represent mean-field ICA [20]

Table 1: MSE of the recovered mixing matrix by different methods on synthetic OICA data.

| Methods | p=2, d=4 | p=3, d=6 | p=4, d=8 | p=5, d=10 |
|---------|----------|----------|----------|-----------|
| RICA | 2.26e-2 | 1.54e-2 | 9.03e-3 | 7.54e-3 |
| MFICA_Gauss | 4.54e-2 | 2.45e-2 | 4.21e-2 | 3.18e-2 |
| MFICA_MoG | 2.38e-2 | 9.17e-3 | 2.43e-2 | 1.04e-2 |
| NG-EM | 1.82e-2 | 6.56e-3 | 1.21e-2 | 6.34e-3 |
| LFOICA | **4.61e-3** | **5.95e-3** | **6.96e-3** | **5.92e-3** |

Table 2: MSE of the recovered causal adjacency matrix by LFOICA and NG-EM.

| Methods | MSE | | | Time (seconds) | | |
|---------|-----|-----|------|----------------|-----|------|
| | n=5 | n=7 | n=50 | n=5 | n=7 | n=50 |
| LFOICA | **1.04e-3** | **5.79e-3** | **1.81e-2** | **75.01** | **76.44** | **1219.34** |
| NG-EM | 6.98e-3 | 9.85e-3 | - | 1826.60 | 4032.54 | - |

with the prior distribution of ICs set to the Gaussian and the mixture of Gaussians respectively. NG-EM denotes the EM-based ICA [15]. $p$ is the number of mixtures, and $d$ is the number of ICs. For each algorithm, we conduct experiments in 4 cases (with $[p = 2, d = 4]$, $[p = 3, d = 6]$, $[p = 4, d = 8]$, and $[p = 5, d = 10]$). Each experiment is repeated 10 times with randomly generated data and the results are averaged. As we can see, our LFOICA achieves best result (smallest error) compared with the others. We also compare the distribution of the recovered components by LFOICA with the ground-truth, the result can be found in Section 2.2 of Supplementary Material.

### 4.2 Recovering Causal Relation from Causal Model with Measurement Error

**Synthetic Data** We generate data with measurement error, and the details about the generating process can be found in section 3.1 of Supplementary Material. NG-EM [15] is a causal discovery algorithm as an extension of EM-based OICA method. Table 2 compares the MSE between the ground-truth causal adjacency matrix and those estimated by NG-EM and our LFOICA. The synthetic data we used contains 5000 data points. We test 3 cases where the number of variables $n$ is 5, 10, and 50 respectively. Each experiment is repeated 10 times with random generated data and the results are averaged. As we can see from the table, LFOICA performs better than NG-EM, with smaller estimation error. We also compare the time taken by the two methods with the same number of iterations. As can be seen, NG-EM is much more time consuming than LFOICA (because EM needs to calculate the posterior). We found that when $n > 7$, NG-EM fails to obtain any results because it runs out of memory, while LFOICA can still obtain reasonable result. So no results of NG-EM is given in the table for $n = 50$. These experiments show that besides the efficacy, LFOICA is computationally much more efficient and uses less space than NG-EM as well.

**Real Data** We apply LFOICA to Sachs's data [41] with 11 variables. Sachs's data is a record of various cellular protein concentrations under a variety of exogenous chemical inputs and, inevitably, one can imagine that there is much measurement error in the data because of the measureing process. Here we visualize the causal diagram estimated by LFOICA and the ground-truth in Figure 3(a) and 3(d). The estimated causal adjacency matrix by LFOICA can be found in section 3.2 of Supplementary Material. For comparison, we also visualize the causal diagrams estimated by NG-EM and the corresponding ground-truth in Figure 3(b) and 3(e). To demonstrate the fact that regular causal discovery algorithm cannot properly estimate the underlying causal relations under measurement error , we further compare the result by a regular causal discovery algorithm called Linear, Non-Gaussian Model (LiNG) [42] in Figure 3(c). Unlike LiNGAM, LiNG allows feedback in the causal model. We calculate the precision and recall for the output of the three algorithms. The precision are 51.22%, 48.94% and 50.00% for LFOICA, NG-EM and LiNG, and the recall are 55.26%, 60.53% and 23.68% respectively. As we can see, LiNG fails to recover most of the causal directions while LFOICA and NG-EM perform clearly better. This makes an important point that measurement error can lead to misleading results by regular causal discovery algorithms, while OICA-based algorithms such as LFOICA and NG-EM are able to produce better results. Although

the performances of LFOICA and NG-EM are very close, it takes about 48 hours for NG-EM to obtain the result while LFOICA takes only 142.19s, which further demonstrates the remarkable computational efficiency of LFOICA.

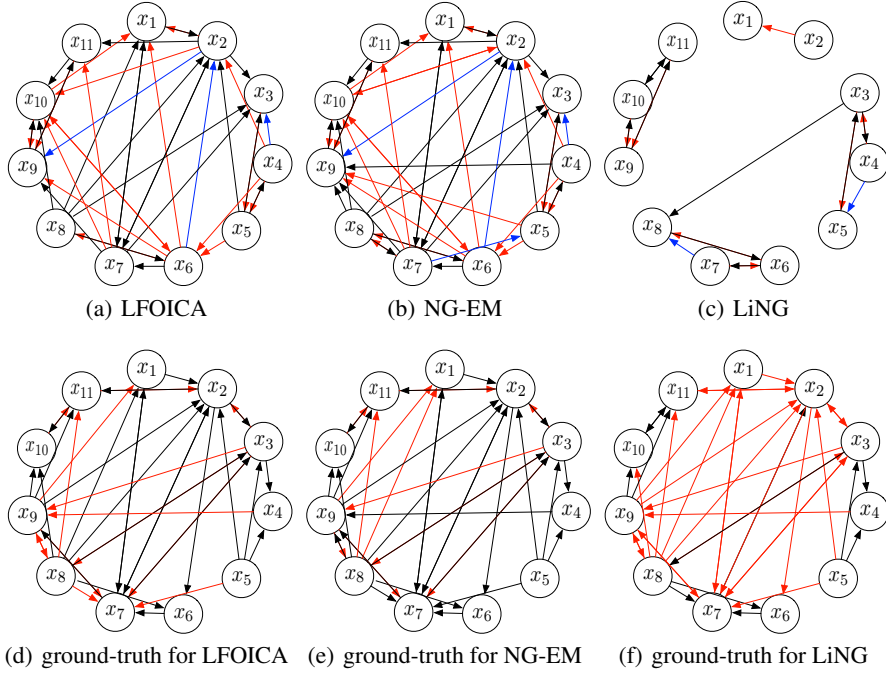

|     |     |     |
| --- | --- | --- |
| (a) LFOICA | (b) NG-EM | (c) LiNG |
| (d) ground-truth for LFOICA | (e) ground-truth for NG-EM | (f) ground-truth for LiNG |

Figure 3: (a)-(c) Causal diagrams by LFOICA, NG-EM and LiNG. (d)-(f) Three ground-truth causal diagrams which are actually the same with the red arrows representing the missing causal directions in the output of the corresponding algorithm. The red arrows in (a)-(c) are falsely discovered causal directions compared with ground-truth. The blue arrows in (a)-(c) are edges with converse causal directions compared with ground-truth.

## 4.3 Recovering Causal Relation from Low-Resolution Time Series Data

We then consider discovery of time-delayed causal relations at the original high frequency (represented by the VAR model) from their subsampled time series. We conduct experiments on both synthetic and real data.

**Synthetic Data**   Following [15], we generate synthetic time series data at the original causal frequency using VAR(1) model described by (6). Details about how the data is generated can be found in section 4.1 of Supplementary Material. NG-EM and NG-MF were first proposed in [15] as extensions of OICA algorithms to discover causal relation from low-resolution data. Table 3 shows the MSE between the ground-truth transition matrix and those estimated by LFOICA-conditional, NG-EM, and NG-MF when number of variables $n = 2$. We conduct experiments when the subsampling factor is set to $k = 2, 3, 4, 5$ and size of dataset $T = 100$ and 300. Each experiment is repeated 10 random replications and the results are averaged. As one can see from Table 3, LFOICA-conditional achieves comparable result as NG-EM and NG-MF [15]. NG-EM has better performance when the number of data points is small ($T = 100$), probably because the MMD distance measure used in LFOICA-conditional may be inaccurate with small number of samples. When the number of data points is larger ($T = 300$), LFOICA-conditional obtains the best results. We also conduct experiment when $n$ is larger ($n = 5$). The result can be found in Section 4.2 of Supplementary Material; again, LFOICA-conditional gives more accurate results and it is computationally much more efficient.

**Real Data**   Here we use Temperature Ozone Data [43], which corresponds to the 49th, 50th, and 51st causal-effect pairs in the database. These three temperature ozone pairs are taken at three

Table 3: MSE of the recovered transition matrix by different methods on synthetic subsampled data.

| Methods | n=2 | | | | | | | |
| | T=100 | | | | T=300 | | | |
| | k=2 | k=3 | k=4 | k=5 | k=2 | k=3 | k=4 | k=5 |
| --- | --- | --- | --- | --- | --- | --- | --- | --- |
| LFOICA-conditional | 7.25e-3 | 7.88e-3 | **8.45e-3** | **9.00e-3** | **1.12e-3** | **3.87e-3** | **4.07e-3** | **6.23e-3** |
| NG-EM | **6.50e-3** | **7.32e-3** | 1.02e-2 | 1.04e-2 | 7.24e-3 | 9.11e-3 | 9.54e-3 | 9.98e-3 |
| NG-MF | 9.09e-3 | 9.89e-3 | 1.24e-2 | 2.19e-2 | 8.46e-3 | 8.76e-3 | 1.01e-2 | 2.20e-2 |

different places in 2009. Each pair of data contains two variables, ozone and temperature, with the ground-truth causal direction temperature $\longrightarrow$ ozone. To demonstrate the result when $n = 2$, we use the 50th pair as in [15]. The optimal subsampling factor $k$ can be determined using the method of cross-validation on the log-likelihood of the models. Here we use $k = 2$ according to [15]. The estimated transition matrix $\mathbf{C} = \left[ \begin{smallmatrix} 0.9310 & 0.1295 \\ -0.0017 & 0.9996 \end{smallmatrix} \right]$ (the first variable is ozone and the second is temperature in the matrix). from which we can clearly find the causal direction from temperature to ozone. We also conduct experiments when $n = 6$. The result can be found in section 4.3 in Supplementary Material.

## 5 Conclusion

In this paper, we proposed a Likelihood-Free Ovecomplete ICA model (LFOICA), which does not require parametric assumptions on the distributions of the independent sources. By generating the sources using neural networks and directly matching the generated data and real data with some distance measure other than Kullback-Leibler divergence, LFOICA can efficiently learn the mixing matrix via backpropagation. We further demonstrated how LFOICA can be extended to sovle a number causal discovery problems that essentially involve confounders, such as causal discovery from measurement error-contaminated data and low-resolution time series data. Experimental results show that our LFOICA and its extensions enjoy accurate and efficient learning. Compared to previous ones, the resulting causal discovery methods scale much better to rather high-dimensional problems and open the gate to a large number of real applications.

## 6 Acknowledgements

Chenwei Ding and Dacheng Tao would like to acknowledge the support by Australian Research Council Projects FL-170100117 and DP-180103424. Kun Zhang would like to acknowledge the support by National Institutes of Health under Contract No. NIH-1R01EB022858-01, FAINR01EB022858, NIH-1R01LM012087, NIH-5U54HG008540-02, and FAIN- U54HG008540, by the United States Air Force under Contract No. FA8650-17-C-7715, and by National Science Foundation EAGER Grant No. IIS-1829681. The National Institutes of Health, the U.S. Air Force, and the National Science Foundation are not responsible for the views reported in this article.

## Footnotes

[1]Code for LFOICA can be found <span style="color:red">here</span>

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
