[Supplementary Material]

# SUPPLEMENTARY MATERIAL FOR "LIKELIHOOD-FREE OVERCOMPLETE ICA AND APPLICATIONS IN CAUSAL DISCOVERY"

## 1 EM Procedure for OICA

In many OICA algorithms, EM procedures are used to maximize the data log likelihood. Here we assume a mixture model for noisy OICA:

$$\mathbf{x} = \mathbf{A}\mathbf{s} + \mathbf{e} \tag{1}$$

Where noise $\mathbf{e}$ is Gaussian distributed with zero mean and covariance matrix $\mathbf{J}$. Each IC $s_i$ in $\mathbf{s}$ is modeled as mixture of Gaussian:

$$P\left(s_i\right) = \sum_{j=1}^{m} \omega_{i,j} \mathcal{N}_{s_i}\left(\mu_{i,j}, \sigma_{i,j}^2\right), \text{ where } \sum_{j=1}^{p} \omega_{i,j} = 1, i = 1, 2, \ldots, n \tag{2}$$

Where $m$ is the number of Gaussian mixing components for each IC. $n$ is the number of ICs. For convenience we introduce an indexing variable $\mathbf{z} = [z_1, \ldots, z_n]$, where each $z_k$ can take a discrete value from 1 to $m$ and represents the state of the mixture for the $k$th IC. We summarize the general EM procedure for OICA in Algorithm 1.

---

**Algorithm 1** EM Procedure for OICA

---

1: Initialize parameters $\mathbf{A}, \mathbf{J}, \boldsymbol{\mu}, \boldsymbol{\sigma}, \boldsymbol{\omega}$
2: E-step: Calculate posterior distribution for latent variables $\mathbf{z}$ and $\mathbf{s}$ using the parameters calculated in previous M-step or from initializaiton.

$$P(\mathbf{z}, \mathbf{s}|\mathbf{x}, \mathbf{A}, \mathbf{J}, \boldsymbol{\omega}, \boldsymbol{\mu}, \boldsymbol{\sigma}) = P(\mathbf{z}|\mathbf{x}, \mathbf{A}, \mathbf{J}, \boldsymbol{\omega}, \boldsymbol{\mu}, \boldsymbol{\sigma}) * P(\mathbf{s}|\mathbf{z}, \mathbf{x}, \mathbf{A}, \mathbf{J}, \boldsymbol{\omega}, \boldsymbol{\mu}, \boldsymbol{\sigma})$$
$$= \tilde{P}_{\mathbf{z}} * \tilde{P}_{\mathbf{s}}$$

3: M-step: Maximize the lower bound of data log likelihood by updating parameters with the posterior calculated in previous E-step.

$$(\mathbf{A}, \mathbf{J}) = \arg \max_{(\mathbf{A}, \mathbf{J})} E\left\{\log P(\mathbf{x}|\mathbf{s}, \mathbf{z}, \mathbf{A}, \mathbf{J})|\tilde{P}_{\mathbf{s}}, \tilde{P}_{\mathbf{z}}\right\}$$
$$(\boldsymbol{\mu}, \boldsymbol{\sigma}) = \arg \max_{\boldsymbol{\sigma}} E\left\{\log P(\mathbf{s}|\mathbf{z}, \boldsymbol{\omega}, \boldsymbol{\mu}, \boldsymbol{\sigma})|\tilde{P}_{\mathbf{s}}, \tilde{P}_{\mathbf{z}}\right\}$$
$$\boldsymbol{\omega} = \arg \max_{\boldsymbol{\omega}} E\left\{\log P(\mathbf{z}|\boldsymbol{\omega})|\tilde{P}_{\mathbf{z}}\right\}$$

4: Iterate between E-step and M-step until convergence.

---

## 2 Recovering Mixing Matrix from Synthetic OICA Data

### 2.1 Synthetic Data

Since the identifiability of the OICA model requires non-Gaussian distribution for all the independent sources, we use Laplace distribution with different variance for each source. The sample size is 3000.

## 2.2 The Recovered Independent Components by LFOICA

To further demonstrate the ability of LFOICA framework to recover the ICs, we draw the scatter plot of the first two recovered ICs in the case where the number of mixtures $p = 2$ and the number of components $d = 4$ in Figure 1(a). The ground-truth is shown in Figure 1(b) for comparison. As we can see that the joint distribution of the two ICs with two different super-Gaussian distributions are well-recovered.

Figure 1: (a) Recovered components. (b) Original components

# 3 Recovering Causal Relations from Causal Model with Measurement Error

## 3.1 Synthetic Data

The causal adjacency matrix is randomly generated with each element uniformly sampled from (0.5, 1). The noise $\mathbf{E}$ for the variables in the measurement-error-free causal model should be non-Gaussian distributed due to the requirement of identifiability [1]. So we use MoG. For each variable, its noise $\tilde{\mathbf{E}}_i$ is composed of two Gaussian components, the first Gaussian component is distributed as $\mathcal{N}(0, 0.01)$, the second Gaussian component is distributed as $\mathcal{N}(0, 1)$, the mixing proportions are set to $0.8$ and $0.2$ for the two Gaussian components. The measurement error $\tilde{E}_i$ for each variable is Gaussian distributed as $\mathcal{N}(0, 0.1)$. The sample size is 5000.

## 3.2 Results of LFOICA on Sachs's Data

The estimated causal adjacency matrix is

$$
\mathbf{B} = \begin{bmatrix}
0 & 0.42 & 0 & 0 & 0 & 0.04 & -0.1 & 0.01 & 0 & -0.06 & 0 \\
0.48 & 0 & 0 & 0.02 & -0.1 & 0.54 & -0.37 & -0.09 & 0 & 0 & 0 \\
0 & 0.07 & 0 & 0.09 & -0.1 & 0 & 0.07 & -0.02 & 0 & 0 & 0 \\
0 & 0 & 0 & 0 & -1 & 0 & 0 & 0 & 0 & 0 & 0 \\
0 & 0 & 0.26 & 0.42 & 0 & 0 & 0 & 0 & 0 & 0 & 0 \\
0 & 0 & 0 & 0.02 & -0.11 & 0 & 0 & 0.44 & 0 & -0.06 & 0 \\
-0.1 & 0.42 & 0 & 0 & 0 & 0.92 & 0 & 0 & 0 & 0 & 0 \\
0 & 0 & 0 & 0 & 0 & -0.58 & 0 & 0 & 0 & 0 & 0 \\
0 & -0.07 & 0 & 0 & 0 & 0.17 & -0.22 & 0 & 0 & 0.96 & -0.17 \\
0 & 0.09 & 0 & 0 & 0 & -0.38 & 0.42 & -0.01 & -0.14 & 0 & -0.35 \\
0 & -0.02 & 0 & 0 & 0 & 0 & -0.01 & 0 & 1.03 & 0 & 0
\end{bmatrix}
$$

# 4 Recovering Causal Relation from Subsampled Time Series Data

See Algorithm 2 for detailed description of the LFOICA-conditional algorithm for subsampled data.

**Algorithm 2** LFOICA-conditional For Subsampled Data
---
1: Get a minibatch of i.i.d samples $\mathbf{z}$ from Gaussian noise distribution.
2: Mapping $\mathbf{z}$ using parameterized $f_{\boldsymbol{\theta}}$, get non-Gaussian noise $\hat{\tilde{\mathbf{e}}}_{t+1}$.
3: Get a minibatch of samples $\tilde{\mathbf{x}}$ from observed dataset $(\tilde{\mathbf{x}}_1, \tilde{\mathbf{x}}_2, ..., \tilde{\mathbf{x}}_N)$ as conditions. $N$ is the total number of data points.
4: For each $\tilde{x}_t$ in the minibatch obtained in step 3, take the corresponding data of next time step $\tilde{\mathbf{x}}_{t+1}$ from $(\tilde{\mathbf{x}}_1, \tilde{\mathbf{x}}_2, ..., \tilde{\mathbf{x}}_N)$ as targets.
5: Generator takes non-Gaussian noise $\hat{\tilde{\mathbf{e}}}_{t+1}$ and conditions $\tilde{x}_t$ as inputs and generate target data $\hat{\tilde{\mathbf{x}}}_{t+1} = \mathbf{C}^k \tilde{x}_t + \mathbf{L}\hat{\tilde{\mathbf{e}}}_{t+1}$, where $\mathbf{L} = [\mathbf{I}, \mathbf{C}, \mathbf{C}^2, ..., \mathbf{C}^{k-1}]$.
6: Update $\mathbf{C}$ and $\boldsymbol{\theta}$ by minimizing the empirical estimate of distributional distance between $p(\hat{\tilde{\mathbf{x}}}_{t+1}|\tilde{\mathbf{x}}_t)$ and $p(\tilde{\mathbf{x}}_{t+1}|\tilde{\mathbf{x}}_t)$ on the minibatch.
7: Repeat step 1 to step 6 until max iterations reached.
---

Table 1: MSE of the recovered transition matrix by different methods on synthetic subsampled data when $n = 5$.

| Methods | n=5 | | | | | | | |
|---|---|---|---|---|---|---|---|---|
| | T=100 | | | | T=300 | | | |
| | k=2 | k=3 | k=4 | k=5 | k=2 | k=3 | k=4 | k=5 |
| LFOICA-conditional | **7.84e-3** | **8.27e-3** | **8.84e-3** | **9.36e-3** | **1.07e-3** | **5.40e-3** | **5.31e-3** | **7.51e-3** |
| NG-EM | 1.50e-2 | 2.32e-2 | 2.02e-2 | 3.04e-2 | 1.24e-2 | 2.11e-2 | 2.54e-2 | 2.98e-2 |
| NG-MF | 3.09e-2 | 3.89e-2 | 3.24e-2 | 4.19e-2 | 2.46e-2 | 3.76e-2 | 3.01e-2 | 4.20e-2 |

### 4.1 Synthetic Data

The transition matrix $\mathbf{C}$ is generated randomly with its diagonal elements uniformly sampled from $(0.5, 1)$ and its non-diagonal elements uniformly sampled from $(0, 0.5)$. This is based on the fact that in most cases the inter-variable temporal causal relation is stronger than that of cross-variable. To make the noise non-Gaussian distributed, we model the noise distribution for each variable as MoG with two components: the first component is distrubuted as $\mathcal{N}(0, 0.1)$ and the second component is distributed as $\mathcal{N}(0, 4)$. Since one of the assumptions for the model to be identifiable with the permutation and scaling indeterminacies eliminated is that the noise distributions for different variables are different non-Gaussian distributions, we use different mixing proportions to mix the two Gaussian components for the noise distributions of different variables. Low-resolution observations are obtained by subsampling the high-resolution data using subsampling factor $k$.

### 4.2 Results on Synthetic Subsampled Data When $n = 5$

We also conduct experiments when $n$ is larger ($n = 5$) in Table 1. As we can see, the accuracy of NG-EM and NG-MF drop rapidly when the number of variables $n$ gets larger while LFOICA-conditional can still obtain reasonable results.

### 4.3 Results on Real Data When $n = 6$

Since the 3 pairs of temperature ozone data are measured in different places, and each of the 3 data pairs contains two variables, we simply concatenate the 3 pairs together to form a new dataset containing 6 variables $(x_1, x_2, ..., x_6)$. The corresponding ground truth causal relation is $x_2 \longrightarrow x_1, x_4 \longrightarrow x_3, x_6 \longrightarrow x_5$. Similarly, we use $k = 2$ according to cross-validation on the log-likelihood of the models [2]. The estimated transition matrix is

$$\mathbf{C} = \begin{bmatrix} 0.8717 & 0.4091 & -0.0126 & -0.0140 & 0.0065 & -0.0586 \\ 0.0180 & 0.7093 & -0.0224 & 0.0866 & 0.0039 & 0.0972 \\ 0.0365 & -0.0008 & 0.8587 & 0.2427 & 0.0626 & 0.0160 \\ -0.0233 & 0.0581 & 0.0323 & 0.6447 & -0.0108 & 0.0840 \\ 0.0188 & 0.0336 & 0.0774 & -0.0552 & 0.8553 & -0.7522 \\ -0.0338 & 0.0730 & 0.0245 & 0.0718 & -0.0013 & 0.5782 \end{bmatrix}$$

Clearly, $C_{12}, C_{34}, C_{56}$ are much larger than others (ignore the elements on the diagonal), which suggests the causal directions are correctly recovered.