[Reviews · NeurIPS 2019]

Reviewer 1



Overcomplete ICA (more sources than data) often becomes feasible after making a parametric assumption on the distribution of sources to make the computation of the likelihood feasible. The authors have proposed a method to estimate the mixing matrix without computing the likelihood. The proposed method is minimizing a distributional distance (MMD) between the generated and observed data when each source is produced by a nonlinear transformation of an independent noise. The generation procedure makes sure that sources are independent. The mixing matrix and the parameters of the generator of each source distribution are learned together. Challenges: Identifiability: The authors proposed the use of MoG as a parametric model for sources when the data is scarce. Such method has been extensively studied by [1] and [2] It is not clear from the paper under which circumstances the proposed algorithm converges and if it converges to the true source distributions and true mixing matrix. The main advantage of ICA method compared with Deep generative or inference methods is identifiability. Can you argue under which conditions the method become identifiable? Regarding the sparsity regularizer, sparse coding has been extensively used to approach ICA and Overcomplete ICA problems [3, 4]. The use of GAN-based methods to solve nonlinear ICA problem was studied in [5]. Given these previous work, I think the proposed method lacks sufficient novelty for acceptance. [1] Choudrey, Rizwan A., and Stephen J. Roberts. "Variational mixture of Bayesian independent component analyzers." Neural Computation 15.1 (2003): 213-252. [2] Mehrjou, Arash, Reshad Hosseini, and Babak Nadjar Araabi. "Mixture of ICAs model for natural images solved by manifold optimization method." 2015 7th Conference on Information and Knowledge Technology (IKT). IEEE, 2015. [3] Olshausen, Bruno A., and David J. Field. "Emergence of simple-cell receptive field properties by learning a sparse code for natural images." Nature 381.6583 (1996): 607. [4] Doi, Eizaburo, and Michael S. Lewicki. "Sparse coding of natural images using an overcomplete set of limited capacity units." Advances in neural information processing systems. 2005. [5] LEARNING INDEPENDENT FEATURES WITH ADVERSARIAL NETS FOR NON-LINEAR ICA Update: Thanks to the authors for providing a detailed answer to my questions. Even though some of my concerns still remain unsolved, I'd like to increase my score from 4 to 6.

Reviewer 2



Independent component analysis (ICA) is a tool for statistical data analysis and signal processing that is able to decompose multivariate signals into their underlying source components. A particularly interesting variant of the classical ICA is obtained by assuming more sources than sensors, that is the overcomplete ICA paradigm. In that case the sources can not be uniquely recovered even if the mixing matrix is known. The overcomplete ICA problem has been solved by assuming some parametric probabilistic models. In that work, a methodology that does not require any parametric assumption on the distribution of the independent sources is proposed. The idea is to learn the mixing matrix by using a generator that allow to draw sample easily. A MLP generator model with standard Gaussian input is learned by minimizing the Maximum Mean Discrepancy (MMD). That is very relevant and offers quite a lot of promising perspectives. The proposed methodology and its application in causal discovery will have an important impact and should be published in NeurIPS 2019 proceedings. UPDATE AFTER THE REBUTTAL After reading all the reviews and the authors' feedback, I maintain my opinion: that contribution is very convincing and deserves to be accepted.

Reviewer 3



The authors present a method for training overcomplete generative ICA models using a GAN approach with no posterior or likelihood calculation. Overall, the methods is clearly described and a simple method to use GANs to perform OICA. The evaluations are clear if somewhat limited in scope. The generative model does not have a corresponding latent inference algorithm, which limits its use. Comparison with other OICA models RICA will likely find degenerate bases when overcomplete [1] and so may not be a great comparison method. Score matching might be a better OICA method [2]? However, score matching ICA models are better described as analysis models and may not be well matched to generative recovery [3]. It would also add to the generality of the method if the performance on a higher dimensional dataset was explored. Natural images patches are commonly used. Does this methods scale up? Small comments Line 89, “some empirical estimator” What estimator are you using? Table 1, what are s.e.m.s? How significant are differences? [1] Livezey, Jesse A., Alejandro F. Bujan, and Friedrich T. Sommer. "Learning overcomplete, low coherence dictionaries with linear inference." arXiv preprint arXiv:1606.03474 (2016). [2] Hyvärinen, Aapo. "Estimation of non-normalized statistical models by score matching." Journal of Machine Learning Research 6.Apr (2005): 695-709. [3] Ophir, Boaz, et al. "Sequential minimal eigenvalues-an approach to analysis dictionary learning." 2011 19th European Signal Processing Conference. IEEE, 2011. POST RESPONSE UPDATES: Author response addressed my concerns.

[Author Response · NeurIPS 2019]



| Methods | p=2, d=4 | | p=3, d=6 | | p=4, d=8 | | p=5, d=10 | |
|---|---|---|---|---|---|---|---|---|
| | std. | p-value | std. | p-value | std. | p-value | std. | p-value |
| RICA | 3.72e-03 | 2.47e-10 | 3.73e-03 | 1.58e-07 | 2.95e-03 | 4.54e-07 | 2.98e-03 | 6.33e-05 |
| MFICA_Gauss | 3.04e-02 | 1.67e-05 | 1.28e-02 | 3.72e-07 | 1.05e-02 | 1.03e-09 | 1.27e-02 | 2.44e-07 |
| MFICA_MoG | 2.08e-03 | 3.01e-11 | 1.07e-02 | 1.14e-03 | 2.08e-02 | 7.95e-03 | 3.97e-03 | 1.74e-05 |
| NG-EM | 1.84e-02 | 4.71e-05 | 1.27e-02 | 1.21e-05 | 7.08e-03 | 2.18e-07 | 7.26e-03 | 2.11e-08 |
| SMICA | 5.83e-02 | 1.59e-01 | 4.12e-03 | 1.30e-03 | 6.01e-02 | 1.97e-02 | 6.58e-03 | 1.07e-05 |
| LFOICA | 1.88e-03 | - | 1.14e-03 | - | 9.07e-04 | - | 1.95e-03 | - |

(a)                                        (b)

Figure 1: (a) Some of the features extracted by LFOICA. (b) std. of the results of each method and the p-value of Welch's $t$-test between each baseline method and our LFOICA.

We would like to thank all reviewers for the constructive comments. We will fully address all the review concerns in the revision. The citation indices below are consistent with those in the main paper.

**To R#1: MoG and sparsity have been studied.** It is true that MoG and sparsity have been extensively studied in existing ICA methods. Aside from the two references suggested by the reviewer, we also cited several representative works in this direction ([14,15,16]). In fact, one major motivation of the proposed LFOICA is that the classical MoG-based ICA methods use maximum likelihoood to estimate the parameters, which involve computationally intractable posterior inference. In our practical considerations, we disscussed in the paper that LFOICA can be further specified with additional constraints, such as MoG and sparsity, in a likelihood-free way that enjoys both statistical and computational efficiency. We will further discuss the suggested references in the "practical consideration" subsection.

**Identifiability and convergence of our model.** The identifiability of the mixing matrix $\mathbf{A}$ in our model ($\mathbf{x} = G_{\mathbf{A},\boldsymbol{\theta}}(\mathbf{z}) = \mathbf{A}[f_{\theta_1}(z_1), \ldots, f_{\theta_d}(z_d)]^{\mathsf{T}}$) follows the identifiability results for OICA [29], which is summarized in the following Theorem. **Theorem 1** Given two OICA models $\mathbf{x} = G_{\mathbf{A},\boldsymbol{\theta}}(\mathbf{z})$ and $\mathbf{x}' = G_{\mathbf{A}',\boldsymbol{\theta}'}(\mathbf{z}')$ that specify distributions $\mathbb{P}(\mathbf{x})$ and $\mathbb{P}(\mathbf{x}')$, respectively. Under the condition that none of the distributions of $f_i(\mathbf{z}_i)$ is Gaussian, if $MMD(\mathbb{P}(\mathbf{x}), \mathbb{P}(\mathbf{x}')) = 0$, then $\mathbf{A}' = \mathbf{A}\mathbf{P}_p\mathbf{S}_p$, where $\mathbf{P}_p$ is a $p \times p$ column permutation matrix and $\mathbf{S}_p$ is a $p \times p$ scaling matrix. The proof is almost the same as that of Theorem 3 in [29], except that in order to guarantee $\mathbb{P}(\mathbf{x}) = \mathbb{P}(\mathbf{x}')$, we use $MMD = 0$ while [29] uses maximum likelihood (KL divergence). Given the identifiability results, the estimated mixing matrix converges to the scaled and permuted version of the true mixing matrix and so does the source distributions. With additional constraints on $\mathbf{A}$, the permutation and scaling ambiguity can be further reduced. The proofs to this are provided in the original papers of the two causal discovery problems evaluated in our paper. **The parameters in our MLPs (*i.e.,*$\theta$) are not identifiable ($\theta \neq \theta'$), but we do not need the identifiability of $\theta$ to perform certain downstream tasks, such as the two causal discovery tasks mentioned in our paper.**

**GAN-based methods to solve nonlinear ICA [21].** Although both our work and [21] use a GAN style approach to solve ICA, they are largely different to each other. **First**, the main purpose of [21] is to recover the ICs instead of how the ICs are mixed (*i.e.,*the mixing matrix). It models the mixing and unmixing procedure implicitly with an encoder-decoder architecture. As a consequence of non-linearity, there is no guarantee for identifiability. In contrast, we concentrate on the mixing matrix estimation for causal discovery purpose. **Second**, the encoder-decoder architecture in [21] cannot be easily extended for OICA because the posterior of ICs cannot be modeled by a deterministic encoder. **Third**, the adversarial training target of LFOICA and [21] are different. While [21] aims at matching the joint distribution and product of marginal distribution of the recovered ICs (this is also how [21] makes the components independent), LFOICA is trained to match the distributions of the generated mixtures and true mixtures. And the estimated ICs by LFOICA are naturally independent because they are generated from independent latent noises with separate networks.

**To R#2: Describe the details of standard overcomplete ICA.** Thanks for the suggestion. We will add an algorithmic description that summarizes maximum likelihood (the EM-type) algorithms for OICA in the Supplementary Material.

**To R#3: The generative model does not have an inference algorithm.** Since the main focus of our work is OICA and it's applications to causal discovery, we put significant effort on mixing matrix estimation instead of latent variable inference. Nevertheless, this is a valuable suggestion and we'll consider extending our LFOICA.

**Comparison to score-matching ICA**. We conduct additional experiments with the suggested score matching ICA (SMICA) under the same conditions. For each experiment setting in Table 1 in the paper, MSE of the recovered mixing matrix obtained by SMICA are 3.52e-2, 6.70e-2, 8.11e-3 and 1.94e-2 respectively, which is not as good as LFOICA.

**Scalability to higher-dimensional data such as images.** LFOICA is originally designed for causal discovery, which is different from image feature extraction. To demonstrate the generality of the method to higher dimensional problems, we apply it to extract features from image patches. The features form a dictionary which can be used with the recovered ICs for image restoration tasks, such as image denoising. Example features are shown in the Figure 1(a). Patch size is 16x16 and the number of atoms is set to 400. We use DCT dictionary for initialization.

**"empirical estimator".** It refers to Maximum Mean Discrepancy (MMD), which is mentioned in line 83.

**"s.e.m.s? How significant?".** Here we interpret "s.e.m.s" as standard deviation of the MSE values in Table 1 of our paper and report it as well as the p-values by conduct Welch's $t$-test between baseline works and our LFOICA in Figure 1(b) (this figure corresponds to Table 1 in our paper). As one can see, the differences are significant enough.

[Meta-Review · NeurIPS 2019]

This paper provides an interesting and working approach for overcomplete ICA that doesn't assume a likelihood. This is a valuable contribution that deserves to be accepted at Neurips.